# Rewiring of Lipid Metabolism in Adipose Tissue Macrophages in Obesity: Impact on Insulin Resistance and Type 2 Diabetes

**DOI:** 10.3390/ijms21155505

**Published:** 2020-07-31

**Authors:** Veronica D. Dahik, Eric Frisdal, Wilfried Le Goff

**Affiliations:** Institute of Cardiometabolism and Nutrition (ICAN), Hôpital de la Pitié, Sorbonne Université, Inserm, UMR_S1166, 75013 Paris, France; veronica.dahik@etu.upmc.fr (V.D.D.); eric.frisdal@sorbonne-universite.fr (E.F.)

**Keywords:** adipose tissue macrophages, metabolic activation, obesity, lipid, inflammation, insulin resistance, type 2 diabetes

## Abstract

Obesity and its two major comorbidities, insulin resistance and type 2 diabetes, represent worldwide health issues whose incidence is predicted to steadily rise in the coming years. Obesity is characterized by an accumulation of fat in metabolic tissues resulting in chronic inflammation. It is now largely accepted that adipose tissue inflammation underlies the etiology of these disorders. Adipose tissue macrophages (ATMs) represent the most enriched immune fraction in hypertrophic, chronically inflamed adipose tissue, and these cells play a key role in diet-induced type 2 diabetes and insulin resistance. ATMs are triggered by the continuous influx of dietary lipids, among other stimuli; however, how these lipids metabolically activate ATM depends on their nature, composition and localization. This review will discuss the fate and molecular programs elicited within obese ATMs by both exogenous and endogenous lipids, as they mediate the inflammatory response and promote or hamper the development of obesity-associated insulin resistance and type 2 diabetes.

## 1. Introduction

Once considered a high-income nation problem, obesity has all but tripled in the last 50 years, reaching pandemic proportions. Estimates indicate that about 641 million adults worldwide are obese (BMI ≥ 30), presenting an alarming mean increase in BMI of 0,61 kg/m^2^ per decade [1]. Obesity is characterized by the accumulation of excess fat arising from a disproportionate energy intake and expenditure. It is the primary risk factor for several pathologic conditions, including nonalcoholic fatty liver disease (NAFLD), insulin resistance (IR), type 2 diabetes (T2D) and cardiovascular disease (CVD), which have also seen their prevalence dramatically rise [2], concomitant with the global rise in the prevalence of obesity.

T2D is a disease met with increased fasting glycaemia due to glucose intolerance and ineffective insulin secretion or signaling. The mechanisms of insulin deficiency are beyond the scope of this review; however, the resulting insulin-resistant state has consequences both systemically and at the tissue level, thus altering the subject’s metabolic status [3]. IR is a feature characteristic of early-onset T2D and may occur in the principal calory repositories including the adipose tissue (AT), skeletal muscle and liver. The central role of AT in maintaining whole-body homeostasis is evidenced during obesity, as ectopic fatty acid (FA) and cholesterol (Chol) accumulation in cell types other than adipocytes leads to metabolic dysfunction. Among the cells capable of ectopic lipid storage, adipose tissue macrophages (ATMs) forming crown-like structures (CLS) have been found to accumulate lipids in obese states [4,5,6]. Indeed, the most salient features of diet-induced AT metabolic inflammation (metaflammation) can be resumed as follows: I) enhanced ATM proliferation and metabolic activation promoting inflammatory cytokine production succeeded by II) monocyte-derived ATM recruitment and foam-like cell formation around apoptotic adipocytes and altered metabolism and III) increased pro-inflammatory signaling creating a vicious cycle of metabolic dysfunction.

Metabolically activated (MMe) ATMs are part of a rather new concept of an activation profile that was introduced by Kratz et al., and distinguishes them both functionally and phenotypically from the M1 and M2 phenotypes [7,8]. More importantly, MMe ATMs present significant changes in their intrinsic metabolism, including lipid mobilization (i.e., lysosomal catabolism, storage within lipid droplets (LDs) and efflux) and utilization (i.e., free fatty acids (FFA)/Chol synthesis and oxidation for energy production or membrane incorporation) [9,10,11]. On this account, the emerging fields of lipidomics and metabolomics have proven key in the study of ATM function in regard to individual lipid species. Particularly, it has brought back interest on the relationship between ATM cellular/membrane lipid composition and inflammatory profile, as it has been evidenced that the inflammatory and adaptive properties of immune cells, including macrophages, are dependent of cellular lipid status [12]. The object of this review is therefore to discuss he current knowledge regarding ATM metabolic activation by dietary lipids. 

## 2. ATM Heterogeneity in Obesity: Activation and Function

The permanent crosstalk that exists between adipocytes and immune cells is of the utmost importance, as several studies have established a causal link between increased excess calory intake and chronic low-grade AT inflammation. Indeed, the activation of the innate immune response during obesity, and more importantly that of ATMs, has proven to be the primary mechanism responsible for the development of insulin resistance (IR) since the discovery that the main source of pro-inflammatory cytokines in AT is in fact ATMs [13,14]. Coherent with this observation, it was shown that ATMs account for the most enriched immune cell in obese AT (from 10% in lean states up to 40% when obese) [15]. Specific depletion of phagocytic cells by intraperitoneal injection of clodronate liposomes protected mice from high fat diet-induced obesity, notably by improving insulin sensitivity, glucose homeostasis and inflammatory status, including decreased plasma tumor necrosis factor-α (TNFα) and increased plasma adiponectin (Ad) [16,17]. Further research selectively targeting ATM genes confirmed their definitive role in obesity-associated metabolic disorders and inflammation by displaying an altered lipid and glucose metabolism as evidenced in lipoprotein lipase (LPL) and TNFα-depleted ATMs [18,19].

Multiple macrophage activation and polarization states have been described in the literature and vary along a continuum, from the classically activated, pro-inflammatory (M1) profile to the alternatively activated, anti-inflammatory (M2) state [20,21]. M1 and M2 macrophages have roles in diet-induced obesity (DIO) and IR, with M2 macrophages being associated with lean, insulin-sensitive states, and M1 macrophages being the predominant subset present in insulin resistant, obese subjects [22,23]. Indeed, reduced bone marrow-derived M1 ATM recruitment, either by monocyte chemoattractant protein-1 (MCP-1) or its receptor C-C chemokine receptor type 2 (CCR2) depletion has been shown to be protective against diet-induced inflammation and IR [24,25]. However, while the M1/M2 dichotomy has been often employed under obese settings thanks to its obvious advantage of reflecting the Th1/Th2 response, it poorly portrays the various challenges associated with DIO. Beyond cytokine production, AT expansion during DIO is accompanied by the appearance of hypoxic zones. Hypoxia provides a means for explaining several metabolic changes seen during obesity, as the hypoxia-inducible transcription factor-1α (*Hif-1α*) is implicated in glucose metabolism, inflammatory cytokine expression, adipocyte hypertrophy and death, among other gene programs [26,27]. Most importantly, it has a role in ATM activation, as myeloid-specific deletion of this transcription factor reduces ATM accumulation and IL-1β production [28]. 

Failure to characterize ATMs from obese subjects, plus shared characteristics between the M1 and M2 phenotype [20] highlighted the possibility of the presence of a new ATM phenotype exclusive to the metabolic disorders seen in obesity. On this account, the treatment of human monocyte-derived macrophages (HMDM) with insulin, glucose and palmitate (PA), all of which are readily available in DIO, generated a polarization state different from the M1 and M2 states that have been referred to as MMe ATMs [7]. Although expressing some canonical M1 and M2 markers, MMe ATMs are not driven by the LPS/TLR4 pathway that ensues in interferon (IFN) production, but are rather induced by the peroxisome proliferator activated receptor-γ (PPARγ) and the autophagy-associated protein p62 [7]. It is interesting to note that PPARγ agonist, rosiglitazone [29] or IL-33 [30] treatment increases M2-like ATM content and reduces macrophage lipid accumulation. This, plus ulterior studies determining that MMe ATMs indeed heighten inflammation, but are also involved in mechanisms for dead adipocyte clearance, such as lysosomal exocytosis [8], would lead us to believe the metabolically active phenotype is most resembling to the M2 state. Accordingly, Silva et al., reported a monocyte-derived Cd11c^+^Cd64^+^ ATM population which significantly increased under a high fat diet (HFD), and was associated with a protective, anti-inflammatory response [31].

Henceforth, numerous researchers have depicted different ATM populations with different metabolic and transcriptomic profiles to the M1 and M2 macrophages, though still all falling under the MMe subtype. For instance, ATMs isolated from visceral AT and epididymal white AT (WAT) from obese individuals and mice, respectively, were shown to express high levels of CD9, as well as genes involved in lipid metabolism and lysosomal functions such as *Cd63, Lpl, Plin2,* and *Lamp2*—the latter two incidentally also upregulated in both landmark studies abovementioned [7,8]. They were also found to accumulate intracellular lipids through a lysosomal-dependent pathway [32]. Coherent with the fact that lysosomal activity is met with sturdy changes in energy metabolism, metabolic activation resulted in the alteration of several metabolic routes. In contrast to LPS-activated M1 macrophages, MMe ATMs revealed increased OXPHOS and a persistent glycolytic rate [27]. Similar observations were made in ATMs (Cd11b^+^ cells in the stromal vascular fraction (SVF)) issued from obese subjects, which displayed a high energy metabolic phenotype with significantly high OXPHOS and glycolysis. These parameters were, however, reduced in ATMs from lean patients, indicative of a low bioenergetic state [33]. Furthermore, Jaitin et al., reported that these lipid-associated ATMs present a large, highly conserved gene expression signature characterized by expression of, among others, *Cd9* and *Cd63, Lipa* (lysosomal function), *Cd36*, *Lpl, Fabp4, Fabp5* (for lipid metabolism) and *Atp6v0b, Atp6v1b2* (involved in OXPHOS). This program was revealed to be under the control of Trem2 and necessary to counteract obesity-related metabolic disorders, as Trem2 deficiency enhanced HFD-induced weight gain, hypercholesterolemia, glucose intolerance and IR [5]. These assessments prompt us to believe that MMe ATMs do not undergo the binary switch associated with the M1/M2 phenotypes, but are hosts to a more dynamic regulation of lipid metabolism which should be taken into consideration for future research.

Lastly, it is interesting to note the spatio-temporal differences of ATM localization and recruitment during lean and obese states. Indeed, M1-like macrophages have been reported to be located in CLS and increased during a HFD, while M2-like ATMs are most often found in interstitial spaces within AT [34]. Recent reports also show the existence of vasculature associated macrophages (VAMs), which seem to be issued from tissue-resident macrophages and tightly associated with blood vessels. Their numbers are, however, more important under lean states [31]. The question as to whether MMe ATMs localize in cluster around dying adipocytes or not is still to be studied; nonetheless, certain similarities with the M2 phenotype, as seen above, suggest they are present in a more scattered pattern.

## 3. Lipid Metabolism in Healthy and Obese AT 

For quite some time now, macrophages have been recognized as capable stocking lipids intracellularly, although most studies were initially conceived in the context of atherosclerosis. A growing body of evidence has nevertheless been able to ascertain that ATMs can also uptake lipids, primarily by buffering circulating Chol and free FA (FFA) (at both early and late stages of overweight/obesity) and dying necrotic-like adipocytes (when advanced obesity). One of the first accounts of lipid laden ATMs in human obesity was observed by Shapiro and coll., who managed to associate “AT foam cell” presence with fasting glucose and insulin levels [4]. This increase in lipid-laden ATMs was maximal at peak adipocyte size [6]. Given that the origin (i.e., circulating vs. adipocyte-issued) of these lipids is determinant of ATM activation, the following section will concentrate on the mechanisms of lipid handling by obese ATMs and its implication in macrophage pro- or anti-inflammatory signaling (Figure 1).

### 3.1. Circulating Lipids: Mechanisms of Uptake into ATMs

#### 3.1.1. FFA

The main tissues for fatty acid (FA) release are muscle and AT, with AT being the preferential target. For this, triglyceride-rich lipoproteins (TRL), including chylomicrons and very low-density lipoproteins (VLDL) are hydrolyzed by tissue-bound LPL and release FFA. These can then be captured by adipocytes and macrophages within the AT for subsequent storage within lipid droplets (LD) (by both adipocytes and ATMs) [32,35] and other intracellular compartments, such as lysosomes (by ATMs) [36]. The primary source of LPL is macrophages, and both its retention at the plasma membrane thereby reducing its bioavailability, or its inhibition by tetrahydrolipstatin treatment has been shown to reduce triglyceride (TG) accumulation in HMDM [37]. Consistently, specific ATM LPL genetic deletion, as performed by Aouadi et al., resulted in reduced lipid storage in macrophages, increased FFA serum levels and glucose intolerance [19].

#### 3.1.2. VLDL/VLDLR

Obesity and obesity-related metabolic disorders are associated with elevated plasma TG levels, high low-density lipoprotein (LDL)-Chol and low high-density lipoprotein (HDL)-Chol contents, with non-HDL levels positively associated with pro-inflammatory ATMs even in healthy subjects [38,39]. Within blood vessel, TG is transported in the form of lipoproteins, including VLDL, which can bind to the VLDL receptor (VLDLR) at the surface of adipocytes and macrophages for TG uptake [40,41]. The expression and proper function of this receptor therefore represent an important target in DIO and diet-induced inflammation, and SNPs of this gene have been found to be associated with BMI and cardiovascular risk in humans [40]. In accordance with this observation, VLDLR overexpression has been shown to increase TG content within ATMs in a DIO mouse model. This was also accompanied by increased M1-like ATM polarization and IR. Improved AT inflammation and insulinemia were, however, restored after the adoptive transfer of VLDLR KO BM to Wt mice even under a HFD [41]. 

### 3.2. Adipocyte-Released Lipids: Mechanism of Uptake into ATMs

#### 3.2.1. Lipolysis-Dependent Lipid Release

Hypertrophic adipocytes are able to release FFA by a classic, lipase-dependent mechanism as an effort to limit further enlargement of the LD that may compromise cell viability. TG hydrolysis and the release of FFA from adipocytes to the circulation or other cells, such as ATMs, are mediated by intracellular lipases adipose triglyceride lipase (ATGL), hormone-sensitive lipase (HSL) and monoacylglycerol lipase (MGL). Adipocyte ATGL deficiency in mice has been shown to decrease ATM infiltration in response to acute lipolysis upon treatment with the β3-adrenergic agonist CL 316,243, as assessed by Cd11c and F4/80 markers. However, a higher ATM infiltration and pro-inflammatory gene expression was detected in AT from adipocyte ATGL-deficient mice upon DIO [42]. Similarly, subjects homozygous for a deletion of the HSL gene display increased MCP-1 levels and ATM infiltration, along with systemic IR and T2D [43]. 

#### 3.2.2. Exosomes

An alternative lipid source for macrophages has been described by numerous authors, whereby lipid-filled exosomes are released by adipocytes and contribute to the pro-inflammatory phenotype of ATM [44,45,46]. This mechanism is independent of lipase activation, as inhibition of ATGL in adipocytes did not alter the distribution of these exosomes, which were incidentally found to increase macrophage TG content by 8-fold in BM-ATMs independently of diglyceride acyltransferase (DGAT)1/2. Moreover, the exosomes were not only able to induce ATM activation and foam-like cell formation, but also lead to ATM-like differentiation from BM progenitors by activating a gene program characteristic of macrophages that reside exclusively in AT [44]. This very same role in ATM differentiation has been found to link exosome release with glucose intolerance and IR [47]. Curiously, a link between the lysosome and exosome pathways has been established, as exosomes were found to deliver TG to lysosomes for FFA release [44].

#### 3.2.3. Exophagy

An active mechanism for lipid release specifically targeting dying, necrotic-like adipocytes around CLS has also recently been described. It entails the formation of a tightly sealed acidic lytic compartment where lysosomal hydrolases, delivered by exocytosis, allow the hydrolysis and subsequent internalization of large amounts of lipids, as well as dead adipocyte fragments [48,49]. This process has been termed exophagy and shares some similarities with autophagy, a widely studied mechanism of lipid clearance. This form of lipid uptake has been shown to lead to foam-like cell formation, as evidenced by LipidTOX labelling [48]. 

### 3.3. Fate of Lipids in ATM

#### 3.3.1. Storage in the Form of LD

Adipocytes and ATMs, among other cells, store fat in the form of LD. LDs are typically composed of a neutral lipid core and sealed off to the cytosol by a set of molecules including membrane phospholipids (PL), perilipins (PLIN), and free Chol [50]. Despite slight differences in the disposition and constitution of the droplet—one large unilocular vesicle formed primarily of TG in adipocytes vs. various smaller vesicles composed of both TG and cholesteryl esters (CE) in ATMs—LD biogenesis and lipid uptake are increased in obesity and contribute to the dysfunctional phenotype observed in both these cell types and associated with DIO [35]. While it was often believed that LDs were the primary—if not the only—form of lipid capture in ATMs, notably those organized in CLS [51], this has been contested by results showing that MMe ATMs present reduced LD formation, as lipids undergo catabolism via a lysosome-related pathway. Most importantly, the inhibition of this pathway increased LD content in macrophages [36]. This form of storage is therefore likely to be preferentially present in CLS ATMs. Likewise, it is mostly found in omental fat rather than subcutaneous AT [4]. 

#### 3.3.2. Lysosome-Mediated Lipid Handling

The contribution of the lysosomal pathway to inflammatory signaling and lipid handling has been the subject of numerous studies. Lipid catabolism, as it occurs in lysosomes, is mainly achieved by the action of LIPA, whose deficiency and overexpression have been shown to alter immune (efferocytosis, T cell and macrophage function) and metabolic (insulin signaling, VLDL metabolism, ATM lipid content, polyunsaturated fatty acid (PUFA) release and lipid mediator synthesis) homeostasis [52,53,54]. In ATMs, the lysosomal program has been associated with the MMe phenotype, as evidenced by the enrichment of lysosome-relates genes *Lipa* and *Lamp2,* among others, in both obese mice [5,32] and humans [36]. As such, its inhibition by chloroquine is accompanied by significantly enhanced lipid accumulation within macrophages and reduced AT lipolysis [36]. Defective lysosomal exocytosis by NAPDH oxidase 2 (Nox2) inhibition also leads to less effective dead adipocyte clearance by MMe and thus contributes to metabolic disorders in DIO [8]. Lysosomal lipolysis has also been associated with the induction of the M2 phenotype in bone marrow derived macrophages (BMDM) and peritoneal macrophages in response to FFA, as assessed by oxygen consumption rate and M2 markers Cd206, Relmα and Cd301 [55]. Interestingly though, while short hairpin RNA treatment targeting *Lipa* did result in impaired M2 polarization [55], hindering lysosomal function has failed to be associated with the classic inflammatory M1 profile [36]. Cell-intrinsic lysosomal lipid handling may therefore represent a means for ATM to counteract classic activation by FFA and would lead to the “mixed” MMe phenotype, which is incidentally characterized by both M1 and M2 markers [7,36].

#### 3.3.3. Autophagy

A complementary mechanism to the lysosomal pathways is thought to be autophagy (also termed lipophagy). This form of lipid mobilization into lysosomes does not conform to a simple role in energy balance, but is also involved in major cellular responses. In AT, for instance, autophagy is necessary for adipogenesis and adipocyte differentiation [56]. In liver and atherosclerotic plaque macrophages, autophagy is required for LD clearance and normal macrophage function [57,58]. In obesity and obesity-associated ATMs, however, autophagy’s role is less clear. In vitro experiments on BM-ATMs and mice under a HFD confirmed autophagy upregulation in DIO; it was nonetheless found dispensable for ATM activation and lipid uptake as evidenced by both *Atg7* deletion and 3-methyladedine treatment [53]. Contradicting results were found using a model of unpolarized RAW macrophages exposed to AT conditioned media plus oleate (OA) as a lipid source. Interrupted autophagy was indeed found to either decrease LD biogenesis and lipid uptake by ATMs when targeted at the early steps, or to support LD formation when targeted at a later stage [35]. These discrepancies could be partly explained by the different lipid substrates (whole AT or OA) to which the macrophages were exposed to. In addition, the inflammatory phenotype of ATMs could also alter lipid efflux efforts by autophagy. It was indeed demonstrated that LPS + IFNγ stimulated macrophages are incapable of further forming and enlarging existing LDs [59]. Lastly, another argument that might allow us to explain these differences has been evoked by Flaherty et al., and involves the existence of a lipid storage site other than LDs that can nonetheless fuse with lysosomes, but does not follow the autophagic route [44]. 

#### 3.3.4. Lipid Efflux/Export

Along with the influx of lipids, their proper clearance is equally important for macrophage function and turns out to be compromised during obesity. Dietary Chol alone is able to further promote ATM accumulation in mice fed a diabetogenic diet, and is positively associated with chronic systemic inflammation, IR and atherosclerosis [60]. This effect is attributed in part to its ability to induce Mcp-1 and Saa3 production by adipocytes, which actively contribute to macrophage recruitment into AT [61]. HDL and Apolipoprotein A-1 (ApoA-1) have also been attributed anti-inflammatory properties in AT [39,61], and while it is worthy to acknowledge their capacity to suppress inflammatory gene expression, HDL themselves can be undergo regulation by inflammatory genes as revealed by identification of SNPs *cis* variants associated with low-HDL [62].

Reducing Chol content in AT through stimulation of Chol efflux may therefore represent an efficient way to alleviate inflammation. For instance, ApoA-1 and HDL exert some of their anti-inflammatory effects in AT in an Abca1/g1-dependent manner by removing Chol from adipocyte membranes in vitro [61]. Intracellular Chol levels are linked to impaired glucose tolerance and IR and as such, Abca1 hematopoietic deletion not only induced monocytosis and increased inflammatory cytokine and macrophage content in both AT and liver, but also enhanced ATM response to saturated FA (SFA). Hematopoietic *Abca1*^-/-^ mice fed a diabetogenic diet supplemented with Chol were thus more insulin resistant [63]. While seemingly logic, these results have been contested by experiments conducted in myeloid *Abca1* KO and Wt mice who developed similar IR, hypercholesterolemia and hepatic steatosis. Although ATM in the SVF from myeloid *Abca1*^-/-^ mice did accumulate more Chol than their Wt counterparts, neither macrophage infiltration nor inflammatory Tnfα, Mcp-1 and Il-1β cytokine expression was different [64]. These differences could be attributed to a number of factors (i.e., hematopoietic vs. myeloid targeting, content and length of the diet, Abca1/g1 levels in AT only vs. in ATM). Regarding Abcg1, its expression has been shown to be significantly enhanced under obese and caloric restriction settings [65,66]. This increase has been revealed to be associated with ATMs, as evidenced by the isolation of F4/80^+^ cells from the SVF of AT obese mice [65]. Particularly, M2 ATM content, assessed by Cd11c^-^Cd206^+^ marker expression, was found to be positively associated with Abcg1 expression. Indeed, its deletion in the myeloid lineage resulted in a 2-fold decrease in M2 macrophages and enhanced total Chol content. This deficiency also hindered macrophage migration in response to Chol loading [66]. Other landmark studies agree that Abca1/g1 macrophage expression is dependent on the ATM phenotype. For instance, Abca1/g1 expression is enhanced in lipid-laden, MMe ATM vs. lean ATM [7]. This is coherent with the observation that while Chol efflux capacity to ApoA-1 was significantly reduced in M1 ATM, it was surprisingly enhanced in primary obese and in vitro polarized MMe ATM [9]. This could be associated with a protective mechanism in which lipid-laden ATMs promote Chol efflux to lower inflammation, but are ultimately overwhelmed by the lipid charge. 

### 3.4. Regulation of Intracellular Lipid Metabolism in ATM

Lipid mobilization both within and outside the cell is a mechanism that strongly depends on three major processes—lipid uptake (either at whole-body level or within the cell), synthesis and clearance or conversion into other metabolites—as these participate in determining the cellular metabolic status. Macrophages, and most importantly ATMs, are capable of regulating their own lipid metabolism essentially through two major families of transcription factors, the sterol regulatory element-binding protein (SREBP) and the liver X receptor (LXR), that tightly control Chol and FA synthesis. We will focus on the mechanisms that drive ATM metabolic dysfunction in obesity. 

Interference with any of the steps involved in Chol metabolism has been associated with altered ATM function. For instance, myeloid deletion of HMG-CoA reductase, the rate limiting enzyme in cholesterol biosynthesis, has been shown to significantly reduce ATM accumulation by 77% in *Hmgcr*^-/-^ mice vs. Wt, despite similar fat mass composition and body weight. This was accompanied by a reduction in Tnfα, Il-1β and Mcp-1 expression, which consistently translated into decreased macrophage chemotaxis and migration into AT, improved glucose tolerance and insulin sensitivity [67]. Similarly, treatment of obese mice with the synthetic LXR agonist GW3965 not only decreased neutrophil migration, but most importantly reduced ATM infiltration—both F4/80^+^Cd11b^+^Cd206^+^ and F4/80^+^Cd11b^+^Cd11c^+^ populations—into visceral AT. Moreover, these changes were associated with enhanced insulin-related gene expression and improved insulin sensitivity in AT and muscle [68]. LXR therefore exerts multiple, non-exclusive roles in inflammation, Chol metabolism and IR.

In contrast to sterol synthesis, FA synthesis or de novo lipogenesis (DNL) is driven by both LXRs and SREBPs, notably the SREBP-1a isoform in macrophages [69]. For this, citrate issued from the TCA cycle is successively transformed into acetyl-CoA and malonyl-CoA, the latter of which can then act as a substrate for the so-called multi-enzymatic complex FA synthase (FAS). The main DNL end product is long chain SFA palmitate (PA; C16:0), which is intrinsically associated with inflammatory pathways that will be detailed later on in this review. Elongation and desaturation steps can then follow for generation of other FA. This process is achieved by different families of enzymes (for details, see Section 4.3.2), which are also under the control of SREBP and LXR. As for Chol synthesis, hindering any of the steps involved in DNL leads to an altered ATM phenotype and lipid metabolism. TCA cycle disruption and corresponding citrate accumulation in LPS-challenged BMDM drives M1-like polarization by overproduction of oxidative stress-associated metabolites such as NO [70]. Myeloid deletion of FAS has shown beneficial effects on glucose tolerance and IR by reducing ATM recruitment to obese AT and pro-inflammatory cytokine expression [71]. In addition, DNL has been found to be required for monocyte differentiation into macrophages [72].

DNL allows the transformation of excess glucose in FA, and as such is upregulated by both insulin and SFA. In regard to insulin, it is able to act at the transcriptional and post-transcriptional levels, the latter consisting mainly of inhibiting proteolytic cleavage and translocation of SREBP1 to the nucleus [73]. At the transcriptional level, insulin has been found to activate mTORC, which is incidentally involved in SREBP-1a cleavage and allows proper phagocytosis downstream of the TLR4 pathway in BMDMs [69]. On this account, TLR4 has been revealed to alter FA synthesis in a biphasic, time-dependent regulation of SREBP-1 expression. Activation of the toll-like receptor TLR resulted in an early (T0 to T6 h) activation phase with enhanced inflammatory cytokine production and impaired FA synthesis—mainly PUFA—by inhibition of LXR and LXR-dependent genes; while the late (T12–T24 h) resolution phase was characterized by upregulation of SREBP-1 and its target genes, which contributed to anti-inflammatory mediator production and counteracted NF-κB signaling [74]. LXR is equally involved in monounsaturated fatty acids (MUFA) and PUFA synthesis, as it upregulates the expression of enzymes necessary for UFA production, such as ACSL3, FADS 1 and 2 and ELOVL5, both in vivo and in vitro in human macrophages and murine foam cells. This effect is achieved in a SREBP-1-dependent manner, as well as by LXR itself in a more direct manner [75]. This lipogenic action on PUFA provides another link, besides Chol efflux and inhibition of its synthesis, for LXR anti-inflammatory, insulin sensitizing effects. In fact, LXR derepression upon NCoR deletion specifically in macrophages contributed to an insulin-sensitive phenotype in all major insulin-target tissues, reduced FFA serum levels and decreased macrophage Tnfα production by impairing TLR response and increasing PUFA production. This was achieved without changes in body weight [76]. These and other mechanisms of Chol and FA metabolism regulation by LXRs have been reviewed in [77,78]

## 4. Membrane Dynamics and Inflammatory Signaling in ATM

A distinct process by which FA and Chol, either captured or newly synthesized, alter ATM inflammatory response in the context of obesity is through their major role as membrane components. Although Chol does not enter the biosynthesis of other lipid classes, FA can be metabolized into a large range of lipid molecules involved in the metabolic activation of ATM with consequences on AT inflammation and IR (Figure 2). Membrane composition and optimal membrane properties (i.e., rigidity vs. fluidity) are crucial for a number of cellular processes such as glucose transport [79], immune cell function and inflammatory signaling [11,71,80,81], AT expansion [82] and even insulin signaling [83,84], all of which are compromised during DIO. Membrane dynamics are therefore of particular interest in the study of diabetes-promoting IR and contribute to this “membrane-centric” view of T2D [85]. 

### 4.1. Altered Chol ATM Content and Membrane Signaling

ATMs from obese mice are characterized by an accumulation of free Chol, which was proposed to contribute to lipid-induced cytotoxicity during DIO [29]. Chol implication in inflammatory processes as portrayed above is in part due to its key role in the formation of lipid rafts which are dynamic microdomains (10–200μm) that allow selective recruitment of other lipids and proteins necessary for intracellular signaling [86]. These rafts have been found to favor the clustering and stabilization of the TLR machinery (i.e., co-receptors CD14 and CD36 and co-adaptor protein for MyD88, TIRAP) [81]. Moreover, given the presence of a Chol recognition amino-acid consensus sequence in the structure of TLR, TLR localization at the membrane and downstream signaling seems dependent on this sterol. Chol loading of both plasma and endosomal membranes in macrophages enhanced TLR4-dependent p38 activation [87]. This effect on inflammation was abrogated by Chol depletion of detergent-resistant membrane domains by Abca1 and β-cyclodextrin, as it disrupts MyD88 recruitment to the membrane; such a mechanism being stimulated by LXR agonists [88]. Similarly, Abcg1-deficient human macrophages and myeloid cells present increased Chol plasma membrane content, which correlates with abnormal LPL retention within lipid rafts and hindered activity, thereby abolishing TG accumulation under VLDL stimulation [37]. Furthermore, Wei et al., demonstrated that changes in membrane Chol and lipid raft content through macrophage FAS inhibition alter global membrane order and composition, thus disrupting Rho GTPase-dependent signaling. These changes were associated with reduced inflammation and IR, a phenotype which was rescued by exogenous sterols but not exogenous FA [71]. These observations further validate Chol involvement in ATM metabolic activation; it is nonetheless of key importance that future research clearly discriminates plasma membrane Chol content vs. other organelles (i.e., ER, endosomes membranes), as well as from that stocked in LD, as this distinction is not always (easily) made. Indeed, accumulation of Chol in ER was reported to induce cytotoxicity and apoptosis in macrophages through the activation of the cell death effector CHOP of the unfolded protein response [89].

### 4.2. Altered PL/FA ATM Content and Membrane Signaling

#### 4.2.1. Glycerophospholipids (GPL)

An ever-increasing number of studies has linked altered membrane glycerophospholipid GPL species—mostly phosphatidylcholine (PC) and phosphatidylethanolamine (PE)—and SFA/PUFA content to metabolic disorders. PC:PE ratio disturbance has been implicated in obesity-associated ER stress, inflammation, decreased insulin signaling and steatohepatitis [90,91,92]. Although most of these studies have focused on the effects of different GPL amounts on hepatocyte membranes, more recent data have found a similar relation between altered PC/PE levels and ATM activation. For instance, PC synthesis, assessed by *Pcyt1a* expression, and lysoPC content is increased in ATMs from genetically obese, insulin-resistant mice. Myeloid-specific inhibition of endogenous PC synthesis improves the metabolic phenotype and lowers macrophage inflammatory response to palmitate, without affecting macrophage number. Instead, this protective effect is mediated by reduced PC turnover at the plasma membrane thus allowing the incorporation of anti-inflammatory, membrane-fluidifying PUFAs, which also contribute to reduced ER stress [11]. This remodeling could be mediated by lysophosphatidylcholine acyltransferase 3 (Lpcat3), although no conclusive evidence was found in the study previously mentioned. LPCAT3 is part of the LPLAT family, and its expression is incidentally high in macrophages. Its role for the re-esterification of lysoPC into PC has been associated with altered inflammatory signaling at the plasma membrane, as activation of this enzyme in macrophages decreased activation of the c-Src/JNK pathway in membrane microdomains [93]. This further highlights the importance of membrane saturation degree on macrophage activation in obesity.

In AT, LXR agonist-mediated insulin sensitizing and anti-inflammatory effects in obese mice were correlated with decreased SFA- and MUFA-containing PCs, while an upregulation of PC and some PE species with high PUFA content was seen. TG species containing C16:0 were equally reduced [68]. Significantly increased expression (+80%) of phosphatidylglycerol (PldG) synthase, the principal enzyme involved in PldG synthesis, was also observed in whole AT depots from HFD-fed mice and positively correlated with high plasma PldG levels and *Tlr4* mRNA expression in WAT. Importantly, daily PldG treatment of obese mice seemed to promote M2-like polarization, as evidenced by increases in M2 markers such as CD163 and CD301. PldGs were also able to inhibit AT lipolysis, albeit through indirect mechanisms [94]. Intriguingly, some adipocyte-derived PL species have been revealed to act in a paracrine manner and influence ATM activation state. Saturated lysoPC generated by phospholipase A_2_ (PLA_2)_ activity were able to promote NLRP3 activation in LPS-primed macrophages and contribute to the general insulin resistant profile in adipocyte-specific *Hif-1α* knockdown mice [95]. Lastly, overweight and obese subjects also display increased serum PldG and PE, both of which are correlated with BMI, IL-6 and Ad expression in a positive and negative manner, respectively, suggesting a link between these PL and low grade, metabolic inflammation [94]. These results, along with those observed in mice, suggest that the obesity-associated increase in PldG levels may reflect a protective mechanism by which ATM-mediated inflammation is (attempted to be) controlled.

Obesity is characterized by an increased oxidative stress in adipose tissue in both humans and mice which was associated with the development of metabolic disorders [96]. Interestingly, Nox2, a key enzyme in the production of ROS, is a driver of the inflammatory and adipocyte-clearing properties of MMe ATM, and ablating Nox2 both improves and worsens DIO-induced metabolic disorders depending on the duration of high-fat feeding [8]. As for FFA and free Chol, PL-bound FAs are subject to oxidation. Oxidized phospholipids (PLs) (oxPL) are recognized as danger-associated molecules and are therefore linked to certain inflammatory pathways. OxPLs derived from PC have been found to induce an Il-1β-driven hyperinflammatory state in macrophages by promoting both OXPHOS and glycolysis; a phenotype also displayed in vivo in hypercholesterolemic mice after LPS administration [97]. OxPL have thus been implicated in steatohepatitis [98], atherosclerosis [99,100] and more recently, in obesity and diet-induced ATM activation [33]. Indeed, Cd11b^+^ cells isolated from the SVF of HFD-fed mice saw their oxPL content double at 9 weeks of HFD. CLS-localized ATM (marked Cx3cr1^+^F4/80^hi^Cd11c^+^Cd206^+^) were abundant in different oxPLs, specifically full-length species which were found to promote Il-1β and Il-6 expression. In contrast, ATM from lean mice (marked Cx3cr1^-^F4/80^lo^Txnrd1^+^HO1^+^) presented higher truncated oxPLs, which were revealed to have antioxidant properties and inhibit mitochondrial respiration in BMDM [33]. Evidence linking oxPL and lipotoxic ceramide production through CerS activation has also been reported [101]. No evidence of oxPL capacity to induce lipid accumulation within ATMs has been observed. Of note, oxPL can equally be found in oxLDL or oxVLDL particles and also accumulate intracellularly in LDs [102].

#### 4.2.2. Sphingolipids (SL)

DIO has also been associated with dysfunctional sphingolipid (SL) metabolism. Among them, ceramides have highly inflammatory, insulin resistant properties that eventually lead to T2D. Ceramide accumulation is observed in plasma and visceral AT from obese women with metabolic syndrome and is accompanied by inflammatory marker expression [103]. Pharmacological inhibition of short chain C16:0 ceramide synthesis correlated with lower weight gain in mice fed a HFD, as well as improved glucose tolerance and insulin sensitivity [104]. The mechanisms guiding ceramide effects are numerous. De novo ceramide synthesis is upregulated by adipose mTORC1 deficiency, which incidentally increases ATM recruitment to CLS, inflammatory cytokine production through NLRP3 activation and IR [105]. Accordingly, ceramide accumulation by VLDLR overexpression in ATM is able to induce M1-like polarization and IR [41]. However, studies conducted with obese mice unable to de novo synthesize ceramides in ATM found contradicting results, as no differences regarding the insulin resistant and NLRP3-dependent inflammatory state were observed. Moreover, this deficiency did not affect M1-like or M2-like polarization, nor adiposity [106]. These discrepancies might be due to cell types—adipocytes vs. myeloid cells—or genes targeted—mTORC1 vs. VLDLR vs. serine palmitoyltransferase long chain 2 (SPTLC2)—and convey the importance of the alternative pathways that might act in a compensatory manner to assure ceramide production.

It is nonetheless unexpected that SPT deficiency did not aggravate the phenotype, as this enzyme also allows the production of sphingomyelin (SM). While the role of SM in ATM-driven inflammation and IR has been less studied, its major contribution to the formation of lipid rafts and LPL bioavailability [107] is expected to induce major changes in inflammatory signaling and lipid storage. This has notably been observed in macrophages where SM synthases 1 and 2 and SPT have been inhibited, and that fail to recruit TLR4 to the membrane, rendering them less sensitive to LPS stimulation [81]. Lastly, altering SM turnover at the membrane by phospholipid transfer protein (PLTP) inhibition in mice has found to be protective against diet-induced weight gain and IR. Since PLTP mediates lipid transfer—mostly SM and Chol—from cell membranes to plasma and vice versa, its deficiency was proven to decrease the levels of these two lipid types in lipid rafts and enhance GLUT4 expression and insulin receptor activity in adipocytes [108]. Given its potential as a target in metabolic disorders, this mechanism might be interesting to explore in ATMs. 

Globally, ATM metabolic activation in DIO is associated with a switch to more lipotoxic lipids. It is important to note that we cannot rule out the contribution of other membrane lipids, such as phosphatidylinositol, phosphatidylserine, phosphatidic acid and plasmalogens. Due to their low abundance in membranes, their effects might be masked by the other lipids. It is thus preferable to study membrane dynamics in relation to individual PL species and nature of the acyl chain. The diet content of ω-3 and ω-6 PUFAs has been found to be of particular importance in determining membrane composition [109] and can thus significantly influence membrane signaling in ATMs.

### 4.3. FA Nature and ATM-Mediated Inflammation

#### 4.3.1. SFA

The inflammatory phenotype of human ATMs is correlated with the FA spectrum of membrane PL, and SFA and PUFA content share a positive and negative association, respectively, with the proportion of these macrophages in obese AT [110]. The SFA to PUFA ratio has also been implicated in ER stress response, and decreased ER saturation and rigidification by inhibition of the de novo PC synthesis pathway, which preferentially selects and binds SFA to PC, has been shown to ameliorate ER stress in BMDMs despite palmitate exposure [11]. Fatty acid-binding protein-4 (Fabp4) acts as an essential lipid chaperone in the enrichment of GPL in SFA and in lipid-induced ER stress in macrophages [111]. Beyond a role in rigidity and fluidity, FAs from membrane PL attain their inflammatory properties by their ability to directly activate inflammatory pathways in macrophages and other cells, or act as precursors for lipid mediators once hydrolyzed from the PL backbone.

SFAs are known for their pro-inflammatory capacity and presence of only single bonds within their acyl chain. SFA with acyl chains up to 16 carbons can be synthetized endogenously, while those containing longer chains are obtained by elongation, with the starting product of stearate (ST; C18:0). They can also be brought by diet, typically animal fats. SFAs are associated with NLRP3 activation; indeed, both PA and ST treatment of human macrophages is able to induce IL-1β secretion. Interestingly, this effect is mediated by rigidification of the plasma membrane, as endogenous SFA induces a significant increase in saturated PC content, which disrupts Na, K-ATPase activity and favors K+ efflux, a mechanism that has been long associated with inflammasome activation. Unsaturated oleate or linoleate co-treatment prevent this outcome [112]. NLRP3 activation is linked to the pro-inflammatory JNK pathway, and JNK activation has been found to exert a role in ATM-mediated IR and development of T2D [113,114], but only under an obesity setting. Mice presenting a macrophage-specific double knockout of *Jnk1* and *Jnk2* displayed a similar metabolic status when fed a normal chow diet. However, when under a HFD, JNK-deficient mice were protected against glucose intolerance and IR, showed decreased ATM infiltration and M1-like polarization [115]. PA can also activate JNK through a mechanism involving the pro-inflammatory p62/NBR1 adaptor proteins in the MAPK pathway, and NBR1 inhibition results in decreased Il-6 and Tnfα production by BMDM. Furthermore, it impairs M1-like polarization in ATMs isolated from HFD-fed obese mice [116]. PA has also been found to induce NF-κB activation in a TLR4 dependent manner, and macrophages from mice lacking Tlr4 revealed blunted inflammatory cytokine production and increased insulin sensitivity [117]. PA has also been shown to induce Tlr4 translocation into lipid rafts in adipocytes [118]. 

Whether PA acts as a direct TLR4 ligand has been extensively challenged in recent years [119]. The first piece of evidence showing PA is not a direct agonist of this receptor was brought in by Lancaster et al., who demonstrated that PA treatment does not induce the conformational changes, dimerization nor endocytosis that characterize TLR4 active state. The absence of TLR4 in murine macrophages, although protective against PA inflammatory effects when used alone, is not sufficient to inhibit PA effects when cells had been previously primed with other TLR agonists [120]. It is interesting to note that this priming of *Tlr4*^-/-^ macrophages was associated with a metabolic reprogramming that rendered them similar to the WT phenotype. These modifications included the upregulation of genes typical of M1-like metabolism and alteration of its lipidome, most notably ceramides, SM, PE plasmalogens and lysoPC/PE plus CE with PA and ST on their acyl chains [120]. Similar observations have been made in studies for PA effect on ceramide production. Indeed, while PA alone was able to induce C16:0 ceramide synthesis and promote Tnfα and Il-1β secretion, this effect was independent of the TLR. In contrast, co-treatment with LPS not only potentiated PA effects, but also turned out to require the TLR4/MyD88 axis. Curiously though, this upregulation of SL murine macrophage content did not involve changes in SPT expression, again hinting at the importance of alternative pathways for ceramide production [121]. In this context, adipose FA binding protein (FABP) was reported to play an important role in ceramide production and cell death in macrophages upon incubation with SFA (PA and stearic acid) [122]. Finally, PA is able to induce the expression of *Hif-1α* in macrophages even under normoxic conditions [28] and to promote pro-inflammatory cytokine expression. This might therefore provide another mechanism, independent of TLR4, for ATM activation during obesity. 

Other SFAs such as lauric acid (LU, C12:0) act through similar mechanisms. LU treatment on RAW macrophages was shown to promote TLR4 recruitment to lipid rafts and the subsequent homodimerization of this receptor, as evidence by GFP, FLAG and FITC labeled constructs of the TLR4/MyD88 pathway [123]. Stearic acid (SA, C10:0) was also shown to induce TLR4 activation, although to a minor degree [124]. This and other mechanisms (i.e., CD36/TLR2 activation in adipocytes and macrophages, IRS-1 phosphorylation, PCG-1 activation) [125,126] contribute to the overall inflammatory and insulin resistant profile seen in DIO and ATM. 

#### 4.3.2. UFA

UFAs, on the other hand, present at least one double bond in their acyl chain and promote an anti-inflammatory environment by inhibiting leukocyte chemotaxis, adhesion molecule expression and production of pro-inflammatory eicosanoids while upregulating that of specialized pro-resolving lipid mediators (SPMs) [127]. When it comes to PUFA, the most prominent species are ω-3 and ω-6 PUFAs. The common precursors for longer, more complex PUFAs are linoleic acid (LA, C18:2n-6) and α-linoleic acid (ALA, C18:3n-3), which have the particularity of being essential FAs that can only be obtained through diet, mostly vegetable fats. Once absorbed, they can undergo a series of elongation and desaturation steps which take place in the ER and involve 3 different enzyme families: ACLS3-4 for FA activation, the ELOVL family for the rate limiting step in elongation and FASD1-2 for desaturation. These enzymes are under the control of SREPB and LXR, as stated above, and thus their transcription is modulated by agonist treatment or derepression of these transcription factors [78]. Their deficiency would lead to impaired long-chain PUFA synthesis, as observed in BMDM from LXR deficient mice, who present decreased levels of ELOVL5, ACSL3 and FASD1 and altered PUFA metabolism—diminution of arachidonic acid and docosahexaenoic acid levels [75]. Elovl5 knockdown mice also presented reductions in ω-3 and ω-6 PUFA levels, while they accumulated C18 PUFAs [128]. 

The most prominent and commonly studied PUFAs are γ-linoleic acid (γ-LA; C18:3n-6), arachidonic acid (AA; C20:4n-6), eicosapentaenoic acid (EPA; C20:5n-3) and docosahexaenoic acid (DHA; C22:6n-3), the latter two sharing strong anti-inflammatory properties. While γ-LA is decreased in the visceral AT of type 2 diabetic, obese women, AA, EPA and DHA are all increased in subcutaneous AT. Most interestingly, AA levels correlate to CD68 expression [129]. Although most AA functions stem from its ability to produce bioactive lipid mediators, AA holds properties innate to itself. AA is the most prevalent PUFA and has been found to disrupt LPS- and SFA-mediated TLR4 activation in macrophages by directly binding to its co-receptor MD2 and decreasing the stimulation of its downstream target NF-κB and MAPK [130]. Contradicting results have, however, arisen from the observation that free AA treatment is able to induce p38 and JNK signaling in human monocytes, as well as to generate a foam-like phenotype by promoting the accumulation of TG, CE and AA itself in LDs [131]. This pool is, however, not used for eicosanoid production [132], but still represents a mechanism that may favor the ATM phenotype distinctive of obesity as lipid-laden monocytes migrate into AT. JNK signaling is associated with apoptotic mechanisms and accordingly, AA treatment of RAW cells caused cell cycle arrest and apoptosis in a JNK-dependent manner [133]. Increased intracellular AA content by altering the Lands cycle also resulted in apoptosis of monocyte-derived macrophages [134].

When it comes to ω-3 PUFA, in vitro and in vivo models display different results. DHA treatment of model membranes resulted in increased membrane fluidity but also an increase in Chol domains, suggesting it is capable of forming distinct Chol-rich or poor domains. On the contrary, EPA, AA and ALA all increased fluidity by reducing Chol domains. EPA notably did so despite temperature and Chol content changes, revealing more consistent effects [135]. However, these results have been contested by observations that EPA treatment of Chol-loaded HMDM does not induce significant changes of the free Chol:GPL ratio. Instead, it participates in foam-like cell formation by reducing ABCA1/G1-mediated Chol efflux and increasing TG content stored in LD [136]. In mice, ω-3 PUFA macrophage production by NCoR depression has been associated with hypoinflammation [76]. Anti-inflammatory M2 polarization and lipid catabolism is promoted in mice under a DHA/EPA-rich diet, where shorter chain FAs were used for β-oxidation while PUFAs were oriented to GPL and SPM production [137]. EPA alone also displays protective, insulin sensitizing effects, as supplementation of this PUFA prevented and even reversed AT inflammation and adipocyte hypertrophy, accounted for by adipocyte size, adipogenesis-related genes, ATM infiltration and FA oxidation (FAO) [138]. Likewise, DHA shares similar effects by promoting M2-like polarization in the AT of obese mice [139]. These protective actions are most likely due to its ability to inhibit NF-κB and NLRP3 inflammasome activation, as well as to target de novo ceramide synthesis and TLR4 recruitment into lipid rafts [123,140,141]. Evidence also suggests that these FAs work in concert with the insulin sensitizing adipokine adiponectin, given that its neutralization augments M1 markers *Il-6, Ccl2, Nos2*, among others, and increases neutral lipid content within ATMs [142]. The discovery of the G protein-coupled receptor (GPCR) GPR120, recently renamed fatty acid receptor 4 (FFA4), as well as DHA and EPA capacity to interact and activate PPARα, represent another means for the anti-inflammatory character of ω-3 PUFA [125,139]. Gpr120 is highly expressed in adipocytes and macrophages, and its deficiency abrogates the anti-diabetic effects that come with ω-3 PUFA supplementation [143,144]. Indeed, DHA and EPA exposure of Gpr120 null mice had no effect on monocyte/macrophage chemotaxis, M1-like and M2-like gene expression as well as overall and tissue-specific insulin sensitivity [143].

Although less studied, MUFAs also display anti-inflammatory properties that can oppose SFA action. A MUFA-enriched diet favored a more insulin sensitive phenotype in mice and humans, as well as reduced IL-1β and NO secretion by macrophages, decreased the number of M1-like ATMs and limited adipocyte hypertrophy [145,146]. In BM-derived macrophages, the ω-9 oleic acid (OA, C18:1n-9) was able to inhibit NLRP3 signaling through the induction of AMPK [145]. This inhibition is consistent with results showing that OA is capable of promoting M2-like polarization in RAW macrophages and mesenteric AT, but not epidydimal AT [106]. The anti-inflammatory ω-7 palmitoleic acid (PM, C16:1n-7) acts independently of AMPK and also PPARs, but is able to lower M1 polarization, cytokine production and ameliorate IR in muscle [114,147]. In accordance with these observations, Cao and et al., showed that PM enrichment in the AT of FABP deficient mice allowed for a significant flux of this FA to the muscle and liver, improving metabolic status and whole-body insulin sensitivity [148].

#### 4.3.3. Eicosanoids and Other Bioactive Lipid Mediators

In addition to directly modifying cytokine production and inflammatory signaling, ω-3 and ω-6 PUFAs are direct precursors for the production of potent bioactive lipid mediators such as eicosanoids and resolvins. AA, being the most abundant ω-6 PUFA, is responsible for the production of prostaglandins (PG), which are generated by the action of cyclooxygenases (COXs), and thromboxanes (TX), leukotrienes (LT) and lipoxins (LX) which are issued by lipoxygenases (LOXs). These metabolites are responsible for most of the bioactivity attributed to AA. EPA and DHA are equal substrates for COXs and LOXs, but also for Cytochrome P450 (CYPs) enzymes which mediate their conversion into protectins (PD), maresins (for DHA) and resolvins (D-series when issued from DHA, E-series when issued from EPA). These 3 families are referred to as SPMs.

Their production is mainly mediated through the action of PLA_2_, which allows the release of different PUFAs from membrane PL [109]. The PLA_2_ superfamily is composed of numerous lipolytic enzymes that are classified according to their induction profile (i.e., cytosolic, secretory, Ca^2+^ independent, among others) [109]. Their implication in the development of metabolic disorders and inflammation has been analyzed in plasma, adipocytes, pancreatic islets and macrophages in both mice and humans. Group X sPLA_2_ has been found to promote PGE_2_ secretion and inhibit glucose stimulated insulin secretion [149]. Similarly, it inhibits LXR activation in macrophages, decreases ABCA1/G1 expression, augments intracellular Chol content in macrophages and promotes inflammatory signaling dependent on TLR4 [149]. PLA2 group V is upregulated in plasma from diabetic subjects and associated SNPs correlate to LDL levels [150]. Moreover, its secretion by hypertrophic adipocytes favorizes M2-like polarization through PGs and other lipid-mediated processes, limits IR in DIO, promotes phagocytosis and intriguingly promotes atherosclerosis [151,152,153]. Other members of the PLA_2_ superfamily also warrant research, notably regarding their possible role in diet-induced ATM metabolic activation. 

LPLATs can also contribute to eicosanoid production as they assure PUFA—preferably AA and EPA—turnover at the *sn-2* position of membrane PL. This has been demonstrated in in vitro and mouse models by altering the expression or activity of the LPCAT3 member. Indeed, while Lpcat3 inhibition in RAW macrophages and shLpcat3-treated mice showed reduced LA- and AA-containing membrane PC and a pro-inflammatory phenotype, its activation reduced the levels of lysoPC and thus limited the amount of free AA that could be used for PGE_2_ production in LPS-primed macrophages [93]. For complementary roles of the PLA_2_ and LPLAT enzymes in membrane dynamics, obesity and obesity-associated metabolic disorders, see [109,154,155]. Lastly, SFA and UFA also control eicosanoid production by regulating the expression of the enzymes mentioned above. Such is notably the case of LU and PA, which were the most potent inducers of COX expression in RAW macrophages. This effect was inhibited by PUFAs such as DHA [124]. Conjugated linoleic acid, on the contrary, displayed an inhibitory effect on COX expression in BMDM and thus resulted in reduced TXA4 and PGE2 synthesis [156]. DHA- and EPA-derived SPMs are characterized by their anti-inflammatory features, while AA-derived eicosanoids are pro- or anti-inflammatory, depending on the context **(**Table 1**)**. Such is the case of PGE2, which has antilipolytic, antifibrogenic, proresolving properties when used as an exogenous stimulus on human WAT explants from obese subjects [157]. It has also been shown to promote an anti-inflammatory ATM phenotype [158]. In contrast, PGE2 production or signaling inhibition has also been shown to be protective against obesity-associated inflammation, as it decreased TNFα, MCP-1 and IL-6 expression by macrophages and AT lipolysis in HFD-fed rodent models and human AT [159]. These discrepancies may notably be due to the specificity of the response, as whole-body inhibition of COX enzymes also downregulates the expression of other prostaglandins (PGs) which might otherwise be responsible for the inflammatory state observed in obesity. This might, however, not be true for cell specific COX inhibition, as obese mice subject to bone marrow transplantation from *Cox1*^-/-^ donors actually showed increased AT inflammation, whereas hepatic inflammation was diminished [160]. Moreover, while PGE2 is the primary ligand for EP3, we cannot rule out the binding of this PG to other receptors, or even other PGs to EP3. Lastly, while PGE2 indeed seemed to promote ATM migration in a model of increased lipolysis after fasting, this was done so without increasing inflammation in AT [161].

The expression of these and other lipid mediators, as well as the enzymes needed for their synthesis is found in human visceral adipose tissue, although their ratio shifts during obesity in favor of AA-derived products [162,164]. Indeed, while the 5-LOX pathway is principally implicated in LT production and, to a lesser extent, resolvin synthesis, its upregulation during obesity is accompanied by increased conversion of AA into LTB4 and LTD4 in obese women with T2D while maintaining resolvin levels below the detection limit [129]. In addition, this pathway has been found to be correlated with LD formation within macrophages which also display higher LTB4 production levels [173]. It is interesting to note, however, that exercise and weight loss promotes the synthesis of other LOX-derived mediators, notably RvD1. This is accompanied by enhanced macrophage phagocytosis [174]. 

Proresolving and pro-inflammatory lipid mediators exert their actions notably through binding to specific GPCRs. RvD1 and LXA4 signal through GPR32/RDV1 and ALX/FPR2, the latter being expressed in murine ATMs and adipocytes as well as human cells [162,163,168]. Unexpectedly, myeloid deletion of *Fpr2* is associated with improved IR, lipid and glucose metabolism as well as lower inflammatory status due to reduced M1-like ATM infiltration [175], meaning other metabolites different to RvD1 and LXA4 are able to bind and more potently induce the FPR2 axis, whose properties seem to depend on the nature of the ligand. RvE1 signals through the receptor ERV1, and its overexpression in myeloid cells holds protective effects on DIO [176]. Interestingly, the chemerin receptor ChemR23 has also been found to bind RvE1 [162]. GPR18/DRV2, LGR6, GPR37 are receptors for RvD2, maresin and PD1, respectively [177,178]. PGE2 signaling is mediated by receptors EP1-4. Among them, EP3 has been shown to be increased in adipocytes of HFD-fed mice and increase adipocyte FFA release [159], while EP4 is induced in ATMs and seems to be implicated in ATM recruitment and lipid buffering [161]. BLT1, also called LTB4r1, is the receptor for LTB4 and is expressed in macrophages. Its deficiency results in higher glucose tolerance, lower blood insulin and FFA levels and improved insulin sensitivity in mice under a HFD. It also translated into lower M1-like ATM migration both in vitro and in vivo and ameliorated inflammatory profile [171].

## 5. Concluding Remarks

Obesity is first and foremost a metabolic disorder characterized by the excessive influx of dietary lipids which orchestrate ATM metabolic activation as a protective means for reducing extracellular and intracellular cytotoxic lipid levels. ATMs are at the crossroad of both lipid metabolism and lipid-mediated inflammation by adopting a specific program for buffering these lipids, a key feature for controlling AT inflammation and the development of IR and T2D. A better understanding of the mechanisms by which lipids shape the ATM phenotype during diet-induced obesity is thus necessary to counteract this pathology, and will most likely need to integrate multi-omic approaches in order to identify interaction networks through which single molecular lipid species are implicated in ATM dysfunction. 

## Figures and Tables

**Figure 1 ijms-21-05505-f001:**
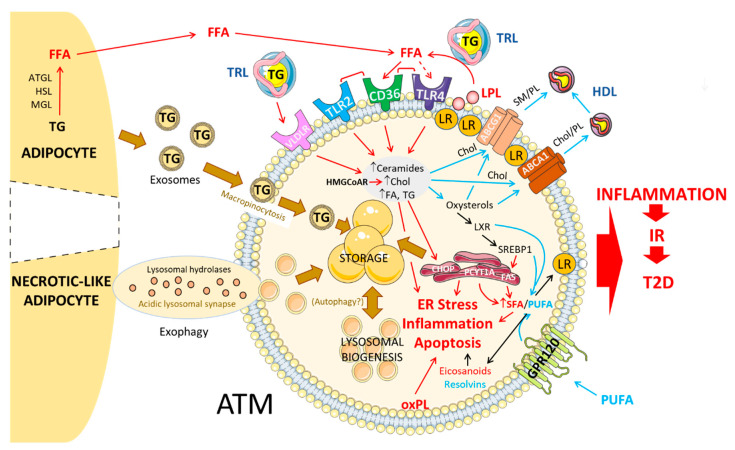
Lipid handling in adipose tissue macrophages in diet-induced obesity.In diet-induced obesity, adipose tissue macrophages are metabolically activated by free fatty acids (FFA) released by lipolysis of triglycerides from adipocytes or from triglyceride-rich lipoproteins by lipoprotein lipase, through Cd36 and toll-like receptors. Accumulation and storage of lipids in adipose tissue macrophages (ATMs) under the form of lipid droplets occur through the uptake of triglyceride-rich lipoproteins by specific receptors such as very low-density lipoprotein (VLDLR) or through the internalization of adipocyte-derived exosomes by macropinocytosis. In crown-like structures, necrotic-like adipocyte clearance by ATM in acidic lysosomal synapses through a mechanism of exophagy also contributes to foam cell formation. As a result, ATMs are characterized by an increased content of triglycerides, free cholesterol, ceramides and fatty acids, which can either be synthesized into other lipids (See Figure 2), be stored in lipid droplets or catabolized through the lysosomal pathway. The accumulation of free cholesterol and sphingolipids such as sphingomyelin into membrane lipid rafts promotes the recruitment of TLR4 at the cell surface and contributes to the activation of inflammatory signaling pathways. ABCA1 and ABCG1 transporters promote the efflux of lipids from ATM to high-density lipoproteins allowing the reduction in lipid raft formation, thus ensuring an optimal lipoprotein lipase activity which participates in foam-like cell formation. Fatty acid synthesis, elongation and desaturation are under the control of liver X receptor (LXR) and sterol regulatory element-binding protein (SREBP1) thus exerting an important role in the relative cellular content of saturated and polyunsaturated fatty acids. Increased cellular levels of free cholesterol and saturated fatty acids trigger endoplasmic reticulum stress, inflammation and apoptosis, which can be alleviated by an enrichment in polyunsaturated fatty acids through mechanisms dependent or independent of the GPR120 receptor. Production of eicosanoids and resolvins from ω-6 and ω-3 polyunsaturated fatty acids, respectively, induces or represses inflammation in ATM. Increased turnover of phosphatidylcholine as well as oxidation of phospholipids also participate in the inflammatory status of ATM. As a whole, the rewiring of the lipid metabolism in ATM promotes adipose tissue inflammation and contributes to the establishment of insulin resistance and type 2 diabetes. ABCA1, ATP-binding cassette A1; ABCG1, ATP-binding cassette G1; ATGL, Adipose triglyceride lipase; Chol, cholesterol; CHOP, CCAAT-enhancer-binding protein homologous protein; FAS, fatty acid synthase; FFA, free fatty acids; GPR120, G protein-coupled receptor 120; HMGCoAR, 3-hydroxy-3-methyl-glutaryl-coenzyme A reductase; HSL, hormone-sensitive lipase; IR, insulin resistance; LPL, lipoprotein lipase; LR, lipid rafts; LXR, liver X receptor; MGL, monoacylglycerol lipase; oxPL, oxidized PL; PCYT1A, phosphate cytidylyltransferase 1; PL, phospholipid; PUFA, polyunsaturated fatty acid; SFA, saturated fatty acid; SM, sphingomyelin; SREBP1, sterol regulatory element-binding protein 1; T2D, type 2 diabetes; TLR, toll-like receptor; TG, triglyceride; TRL, triglyceride-rich lipoproteins; VLDLR, very low-density lipoprotein receptor.

**Figure 2 ijms-21-05505-f002:**
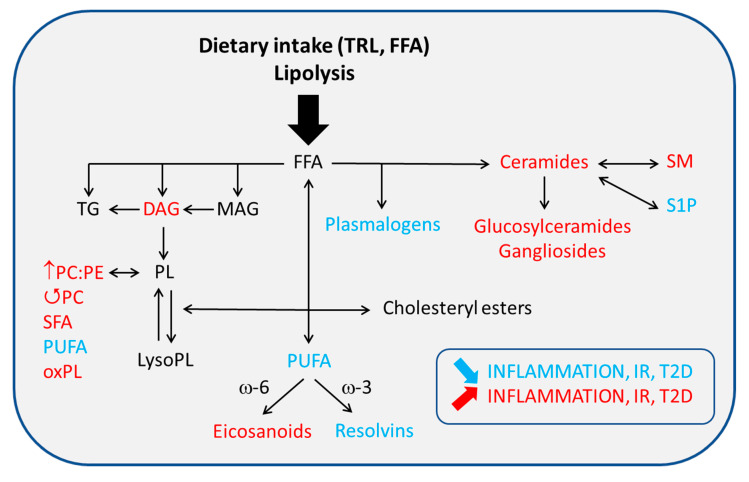
Fate of fatty acids in adipose tissue macrophages. The association of fatty acid (FA)-derived lipids with inflammation, IR and T2D is indicated in blue (protective) or red (deleterious). DAG, diacylglycerol; FFA, free fatty acids; IR, insulin resistance; MAG, monoacylglycerol; oxPL, oxidized PL; PC, phosphatidylcholine; PE, phosphatidylethanolamine; PL, phospholipid; PUFA, polyunsaturated fatty acid; S1P, sphingosine-1-phosphate; SFA, saturated fatty acid; SM, sphingomyelin; T2D, type 2 diabetes; TG, triacylglycerol; TRL, triglyceride-rich lipoproteins; ↺ turnover.

**Table 1 ijms-21-05505-t001:** Principal PUFA-derived lipid mediators in adipose tissue (AT) inflammation and ATM activation during obesity. H: human, M: mouse, R: rat, En: endogenous, Ex: exogenous.

MEDIATOR	PRECURSOR	SPECIES: MODEL	SOURCE	CELL TYPE/TISSUE	REPORTED ACTIONS	REFERENCES
PRORESOLVING MEDIATORS						
RVD1	DHA	M: DIO	En, Ex	ATMs, Peritoneal macrophages	▪ Reduced pro-inflammatory markers (TNFα, IL-6, leptin) and ROS ▪ Induces phagocytosis	[139,162]
		M: *db/db* mice	Ex	ATMs	▪ Improved glucose homeostasis and insulin sensitivity ▪ Increased adiponectin secretion ▪ Decreased CLS formation and ATM M1 polarization	[163]
		M: DIO H: lean and obese	En, Ex	Epididymal AT, BMDM, adipocytes, PBMC	▪ Reduced monocyte/macrophage transadipose migration	[162]
		H: lean and obese ± T2D and THP-1 cells	En, Ex	Omental AT and THP-1 macrophages	▪ Promotes the IL-10/STAT3 pathway at low concentrations ▪ Increases p38 MAPK phosphorylation and IL-1rα production	[164]
RVD2	DHA	M: DIO H: Lean and obese	Ex	Epididymal AT	▪ Reduced pro-inflammatory markers (TNFα, IL-6, IL-12, leptin) ▪ Induced adiponectin secretion ▪ Decreased leptin secretion	[162]
RVE1	EPA	M: *ob/ob* mice	Ex	Epididymal AT	▪ Induced expression of adiponectin, GLUT4, IRS-1 and PPARγ in AT ▪ Reduced monocyte/macrophage transadipose migration	[165]
PD1	DHA	M: *ob/ob* mice	Ex	Epididymal AT	▪ Induced expression of adiponectin	[165]
MARESIN 1	DHA	M: DIO and *ob/ob* mice and H: lean subjects	Ex	Epididymal AT, subcutaneous AT	▪ Ameliorated insulin signaling (AKT phosphorylation) and glucose homeostasis (GLUT4, AMPK expression, blood glucose levels) ▪ Reduced IL-Iβ, TNFα, MCP-1 expression in AT ▪ Attenuated ATM infiltration and reduced the M1:M2 ratio ▪ Induced adiponectin secretion	[166]
LXA4	AA	M: DIO	Ex	Epididymal AT	▪ Induced adiponectin secretion	[162,167]
		M: DIO and 3T3-L1, J774 cells	Ex	ATMs, 3T3-L1 adipocytes, J774 macrophages	▪ Promoted an anti-inflammatory ATM phenotype ▪ Downregulated diet-induced autophagy	[167]
		M: Standard diet and 3T3-L1, J774 cells	Ex	Perigonadal AT and 3T3-L1 adipocytes, J774 macrophages	▪ Decreased IL-6, TNFα and MCP-1 secretion by AT and macrophages ▪ Increased insulin sensitivity by promoting IL-10, GLUT-4 and IRS-1 expression	[168]
PGD2	AA	M: DIO + aP2-Cre/L-PGDS^flox/flox^ mice, DIO and *ob/ob* mice and H: obese subjects	En, Ex	ATM, epididymal AT, subcutaneous AT, BMDM	▪ Decreased weight gain and promoted insulin sensitivity ▪ Altered FA metabolism: decreased SREBP-1, SCD1, FAS ▪ Increased ATGL- and MGL-mediated lipolysis ▪ Altered M1:M2 ratio in favor of anti-inflammatory M2 macrophages	[169,170]
PGE2	AA	H: Lean and obese and 3T3-L1 cells	En, Ex	Omental AT, primary adipocytes, 3T3-L1 adipocytes	▪ Decreased inflammatory and fibrogenic gene expression ▪ Inhibited adipocyte lipolysis ▪ Promoted AT browning	[157]
		M: DIO + CREB^LysMKO^ mice	Ex	BMDM	▪ Mediated M2-like ATM polarization via CREB, thus maintaining insulin sensitivity	[158]
**PRO-INFLAMMATORY MEDIATORS**						
PGE2	AA	H: lean and obese, M: *db/db* mice, R: DIO and 3T3-L1, RAW cells	En	SVF, Primary adipocytes, 3T3-L1 adipocytes, RAW macrophages	▪ Decreased M1-like ATM content and pro-inflammatory cytokine expression by COX or EP3 inhibition ▪ Decreased macrophage migration and responsiveness to hypoxia by COX inhibition ▪ Lower blood glucose levels and higher insulin sensitivity by COX or EP3 inhibition	[159]
LTB4	AA	M: DIO and U937, 3T3-L1 cells	En, Ex	Peritoneal macrophages, 3T3-L1 adipocytes,	▪ Stimulated macrophage chemotaxis ▪ Promoted activation of the NF-κB and JNK pathways in macrophages ▪ Increased TNFα, IL-6, MCP-1, CXCL1 expression ▪ Promoted IR in liver and muscle	[171]
LTD4	AA	M: DIO and *ob/ob* mice	En, Ex	Epididymal AT	▪ Induced NF-κB activity	[172]

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
