# Peer review of "Rewiring of Lipid Metabolism in Adipose Tissue Macrophages in Obesity: Impact on Insulin Resistance and Type 2 Diabetes"

_ijms, 2020, doi:10.3390/ijms21155505_

Round 1

Reviewer 1 Report

This review was written to show lipid metabolism in adipose tissue macrophages (ATMs) in obesity. Inflammation is critical problem in obesity, because it occurs a variety of metabolic diseases. The action and functions of ATMs have been extensively studied, and many papers were published. This review is informative and easy to read. I have no critical comments, but some minor concerns had better be addressed.

  1. In Fig. 1, TG is metabolized by ATGL, HSL, and MGL. HSL and MGL, but not only ATGL had better be also included in the figure.
  2. Triacylglycerol was shown as TG in Fig. 1, but TAG in Fig. 2. It should be unified.

Author Response

We thank this Reviewer for his/her positive comments. The modifications requested by the Reviewer in Figures 1 and 2 have been performed.

Reviewer 2 Report

The review is focused, densely argued, and accurately referenced. It’s basically well written and very enlightening. Dahik et al. described and discussed in depth the importance of dietary lipids in the metabolic activation of ATMs and their role in inducing a pro-inflammatory state implicated in the pathogenesis of obesity-related disorders. I have no major comments, just minor suggestions:

  • Although the focus of this review is to address current knowledge of metabolic activation of ATM by dietary lipids, authors can not avoid considering and briefly discussing other triggers (such as hypoxia, hyperglycemia, adipocytes death, and hyperinsulinemia) other than dietary lipids, which contribute to the modeling of metabolic, inflammatory and functional characteristics (i.e. the phenotype) of ATMs in the obese state.
  • Growing research and studies in immunometabolism and, in particular, in the metabolism of macrophages, provide new therapeutic opportunities for the treatment of inflammatory diseases and cancer. In particular, the metabolic reprogramming of macrophages appears to be an important approach in the fight against diseases with high macrophage involvement, such as cancer, obesity, and atherosclerosis. As I said earlier, several studies have been performed so far that established several approaches aimed at reprogramming the metabolism of macrophages to treat diseases such as obesity and atherosclerosis [Miller et al (2010) Interleukin-33 induces protective effects in the adipose tissue inflammation during obesity in mice; Rombaldova et al (2017) Omega-3 fatty acids promote the use of fatty acids and the production of pro-resolution lipid mediators in alternately activated macrophages of adipose tissue; Furukawa et al (2004) Increased oxidative stress in obesity and its impact on metabolic syndrome; Xu et al (2015) NOTCH reprograms mitochondrial metabolism for activation of proinflammatory macrophages; Jung et al (2016) Chronic repression of mTOR Complex 2 induces dietary-induced changes in the gut microbiota of obese mice; Ricardo-Gonzalez et al (2010) The immune axis IL-4/STAT6 regulates the metabolism of peripheral nutrients and insulin sensitivity]. Since I consider this topic to be particularly interesting, I believe that the authors should address it in their review, briefly describing and discussing some of these studies.
  • Check the references and place them where they are missing from the text.

Author Response

We thank this Reviewer for his/her positive comments and his/her useful suggestions.

We agree with the Reviewer that other triggers (such as hypoxia, hyperglycemia, adipocytes death, and hyperinsulinemia) other than dietary lipids contribute to the modeling of metabolic, inflammatory and functional characteristics (i.e. the phenotype) of ATMs in the obese state. Some of these points were already discussed in the manuscript. However, we have carefully checked the references suggested by the Reviewer and placed the relevant ones in the manuscript. Then, the following references have been inserted in our manuscript : Miller et al. page 3, line 7; Furukawa et al. page 11, line 32, Rombaldova et al. page 14, line 42. Nevertheless, it appears to us that some of the articles mentioned by the Reviewer mainly reported the metabolic activation of cell types other than adipose tissue macrophages which are the purpose of the present review. Thus, the role of the STAT6 by IL-4 axis on insulin sensitivity reported by Ricardo-Gonzalez et al (2010) mainly relied from its action in hepatocytes whereas the NOTCH1 pathways was implicated in the M1 activation of hepatic macrophages (Xu et al., 2015). The beneficial impact of the mTORC2 repression on diet-induced obesity resulted from effects on gut microbiota and intestinal inflammation (Jung et al, 2016) without involving ATM.